# Green Synthesis and Characterization of Zinc Oxide Nanoparticles Using *Eucalyptus globules* and Their Fungicidal Ability Against Pathogenic Fungi of Apple Orchards

**DOI:** 10.3390/biom10030425

**Published:** 2020-03-09

**Authors:** Hilal Ahmad, Krishnan Venugopal, Kalyanaraman Rajagopal, Savitha De Britto, Boregowda Nandini, Hosur Gnanaprakash Pushpalatha, Narasimhamurthy Konappa, Arakere C. Udayashankar, Nagaraja Geetha, Sudisha Jogaiah

**Affiliations:** 1School of Life Sciences, Department of Biotechnology, Vels University, Pallavarm, Chennai 600117, Tamilnadu, India; hhiillaallbiotech@gmail.com (H.A.); 2Department of Biochemistry, Vivekanandha College of Arts & Sciences for Women, Elayampalayam, Tiruchengode 637205, Namakkal Dist., Tamilnadu, India; venuapcas@gmail.com (K.V.); 3Department of Botany, Ramakrishna Mission Vivekananda College, Chennai 600004, Tamilnadu, India; krajagopal@rkvmc.ac.in (K.R.); 4Laboratory of Plant Healthcare and Diagnostics, PG Department of Biotechnology and Microbiology, Karnataka University, Dharwad 580003, Karnataka, India; jsudish@kud.ac.in (S.J.); 5Division of Biological Sciences, School of Science and Technology, University of Goroka, Goroka 441, Papua New Guinea; debrittos@unigoroka.ac.pg (S.D.B.); 6Department of Studies in Biotechnology, University of Mysore, Manasagangotri, Mysuru 570 006, Karnataka, India; bnandini2010@gmail.com (B.N.); n.murthy10@gmail.com (N.K.); ac.uday@gmail.com (A.C.U.); geethabiotech.uom@gmail.com (N.G.); 7Department of Botany, Maharani’s Science College for Women, JLB Road, Mysuru 570 001, Karnataka, India; pushpa_2001@rediffmail.com

**Keywords:** *Eucalyptus globules*, zinc nanoparticles, apple, antifungal activity, FTIR, SEM, TEM, Fruit crops

## Abstract

*Eucalyptus globules* belonging to the Myrtaceae family was explored for the synthesis of zinc oxide nanoparticles and for biological applications. The aqueous extract of the synthesized zinc nanoparticles (ZnNPs) was characterized using UV-visible spectrophotometer, FTIR, SEM and TEM. The aqueous broth was observed to be an efficient reducing agent, leading to the rapid formation of ZnNPs of varied shapes with sizes ranging between 52–70 nm. In addition, antifungal activity of the biosynthesized ZnNPs was evaluated against major phytopathogens of apple orchards. At 100 ppm of ZnNPs, the fungal growth inhibition rate was found to be 76.7% for *Alternaria mali,* followed by 65.4 and 55.2% inhibition rate for *Botryosphaeria dothidea* and *Diplodia seriata*, respectively. The microscopic observations of the treated fungal plates revealed that ZnNPs damages the topography of the fungal hyphal layers leading to a reduced contraction of hyphae. This considerable fungicidal property of ZnNPs against phytopathogenic fungi can have a tremendous impact on exploitation of ZnNPs for fungal pest management and ensure protection in fruit crops.

## 1. Introduction

Nanoparticles are designated as an extension stuck between bulk material and atomic or molecular compositions. The lesser mass and widespread surface to volume ratio of nanoparticles reveal their significant properties with purposeful applicability in the field of sensors, medical, catalysis, optical devices, DNA labeling and drug delivery. Similarly, nanoparticles are focused for their key applications in medical research, re-assign food and agricultural wastes to energy and also other supportive by-products by enzymatic nano-bioprocessing [1], along with further applications in agriculture for the management of phytopathogens by diverse types of nanocides [2,3,4,5]. Among the metal nanoparticles, Zinc nanoparticles (ZnNPs) have shown to be having magnificent applicability in molecular diagnostics, detection and micro-electronics [5], and also in drug delivery and bioimaging probes [6]. They also show exclusive optical and electrical properties [7] due to which, they have various applications in, cosmetics, gas sensors, solar cells, catalysts and ceramics [8], biosensors [9], and solar cells [10]. ZnNPs have also been widely studied for their anti-bacterial and UV-blocking properties [11]. ZnNPs possess an exceptional property against bacterial cellulase, which has been reviewed from time to time [12,13], through to the production of hydrogen peroxide from the surface of ZnNPs [14]. Moreover, the anti-bacterial potential of Zinc oxide nanoparticles was observed to be effective than that of Zinc oxide, essentially classified due to the smaller particles comprising a superior surface-dependent to amount of volume, which exhibits significant anti-bacterial properties [15]. ZnNPs also had substantial toxicity against cancer cells as discovered by various researchers [16,17]. ZnNPs are also considered to be top photocatalysts, which are employed in sanitizing wastewater, and decay or reduce herbicides and pesticides [1].

The commercial routes of ZnNPs preparation includes hydrothermal synthesis [18], electrochemical method [19], mechano-chemical method [20], laser ablation [21], sonochemical [22], polyol method [23], vapor-phase transport method [24], sol-gel method [25], precipitation method [26], microwave technique [27], and by aerosol process [28]. These methods can be adopted for either chemical synthesis or plant-derived synthesis of nanoparticles. For the chemical synthesis of metal nanoparticles, specific external catalysts and synchronized conditions are required, in case of plant-derived nanoparticles catalysts in the form of co-enzymes are secreted by plants which are non-toxic reactants and eco-friendly and the reaction occurs at favorable room temperature conditions. Hence, there is an emerging concern towards the use of plant extracts so called phytosynthesis of nanoparticles due to the presence of plant bio-molecules that can act as capping and reducing agents. These agents, thus, increase the rate of reduction and stabilization of nanoparticles. In addition, the use of plants is an adventitious process over commercial products because it does not involve the long process of growing cell cultures and is more useful for large-scale nanoparticle synthesis. Hence, an increase in the biogenesis of nanoparticles using plant materials is tremendously growing, for instance, the synthesis of ZnNPs by using various plant materials are well documented in *Acalypa indica* and *Corriandrum sativum* [29], *Aloe barbadensis* [30], *Morinda pubescens*, *Passiflora foetida* [31], *Passiflora foetida*, *Hybanthus enneaspermus*, *Aloe barbadensis*, *Parthenium hysterophorus* [32], and milky latex of *Calotropis procera* [33].

Apple (*Malus domestica* Borkh.), a predominant fruit crop of Jammu and Kashmir State, India is affected by many fungal diseases. Recently, Alternaria leaf blotch, stem and branch cankers existed as epidemic diseases, causing extensive damages to the apple trees [34]. These diseases are exhibiting epiphytotic proportions in some pockets of Kashmir valley, India [34], resulting in huge economic losses [35]. 

The present study focuses upon the formation kinetics of ZnNPs by *Eucalyptus globules* as reducing agent. *E. globules*, a rapid growing herb with medicinal properties used in India. It is used in analgesic, anti-inflammatory and anti-pyretic remedies [36]. The essential oils like, Aromandendrene myrtenal, Borneol, Camphene, Carvacrol, Citronellal, Citronellyl acetate, Cryptone-α-terpenyl acetate of *E. globules* are used for medicinal and pharmaceutical purposes [37]. The anti-bacterial, antifungal, anti-oxidative and anti-radical properties of *E. globules* have been well documented [38]. Additionally, elucidation of the fungicidal property of ZnNPs against major phytopathogens of apple orchards carried out in the present study is for the first time. 

## 2. Materials and Methods

### 2.1. Bio-Material Preparation

*Eucalyptus globules* leaves were selected as bio reducing agent for the present study (Figure 1). Fresh leaves of *Eucalyptus globules* were collected within the Jamin Pallavarm Vels University, Chennai, India and identified by Dr. Akhtar, Center of Biodiversity and Taxonomy, Department of Botany, University of Kashmir (India). Plant materials were shade-dried, rinsed in distilled water followed by sterilization with mercuric chloride (0.1%) for 30 sec, followed by five times repeated washes with sterile water and later shade-dried. Dried leaves were powdered using laboratory blender and used for further studies. 15 g of the leaf powder was taken in a beaker containing 200 mL de-ionized water and incubated in a shaker at 80 °C with 1500 rpm for 6 h. The extract was centrifuged at 10,000 rpm for 10 min, and the supernatant was filtered using Whatman No.1 paper to obtain a final volume of 100 mL.

### 2.2. Phytosynthesis of Zinc Nanoparticles

One mM of Zinc nitrate hexahydrate [Zn(NO_3_)_2_6H_2_O] was suspended in 10% plant extract solution in 1:2 ratio with continues stirring, after the mixture was completely dissolved, it was placed for stirring at 150 °C for 2–3 h, and the resultant mixture was cooled at room temperature, and the supernatant was discarded. The resultant solid product (pale white in color) was centrifuged twice at 6000 rpm for 10 min, washed and dried at 80 °C for 5–6 h. The dried powder was kept at room temperature until it changes its color and used for further studies. 

### 2.3. Characterization of Nanoparticles

#### 2.3.1. Detection of Zinc Nanoparticles using UV-Vis Spectrophotometer

The surface resonance plasmon absorption transition of ZnNPs was performed with 1 cm quartz cuvettes using the spectrophotometer (UV-260, Shimadzu Corp. Tokyo, Japan) and the absorbance was read in the wavelength of 200–700 nm.

#### 2.3.2. Characterization of Zinc Nanoparticles

Studies on characterization of *E. globules* ZnNPs were carried out at UGC-DAE-CSR Indore, India. FT-IR spectral analyses were performed using VERTEX-70, Bruker spectroscopy (Optik GmbH, Ettlingen, Germany). The different functional groups present in ZnNPs pellet was checked by treating with KBr and scanning at 400–4000 cm^−1^. The size and shape of the ZnNPs were observed using scanning electron microscopic studies (JSM 5600, JEOL, Tokyo, Japan) by placing the ZnNPs on a copper grid coated with carbon and measured at 80 kV. To obtain the TEM images, ZnNPs were placed on copper grid coated (200-mesh) with carbon and observed using Tecnai G2 20 microscope (FEI, Amsterdam, The Netherlands).

### 2.4. Assessment of Antifungal Activity of Zinc Nanoparticles Against Pathogenic Fungi

#### 2.4.1. Fungal Culture

Three pathogenic fungal species infecting apple orchards viz., *D. seriata, B. dothidea* and *A. mali* were obtained from the Department of Plant Pathology, S.K. University of Agricultural Sciences and Technology, Srinagar, India. The fungal cultures were grown at normal temperature and then further maintained at low temperature in Potato Dextrose Agar (PDA) slants. All fungal growth inhibition experiments were done on PDA media previously treated with test fractions. 

#### 2.4.2. Antifungal Assay

ZnNPs were evaluated against *A. mali, D. seriata* and *B.dothidea* for the estimation of their antifungal activity following the protocol of Jogaiah et al. [39]. A volume of 0.5 ml of each 0.05, 0.10, 0.25, and 1 milligram per milliliter concentrations were aseptically poured into the petriplates loaded with 9.5 mL of molten PDA. For proper dispersal, the media was stirred well at 150 rpm, incubated at room temperature for three days. For the comparative studies, Zinc nitrate solution and *E. globules* extracts were evaluated. An inoculum disc (5 mm) of each test fungi (slashed from a growing mycelium) was inoculated at the middle of the medium and incubated at 24 ± 2 °C for 5–7 days. 

### 2.5. Measurement of Mycelial Inhibition 

The average diameter size of the growing mycelium was measured using micro-scale, data was recorded on the 7th day of incubation, and the percent inhibition of mycelium (PIM) was measured using the formula;
Inhibition of mycelium (%) = (g^c^−g^t^)/g^c^ × 100
where; g^c^= control plates showing the growth of mycelium and g^t^= ZnNPs-treated plates showing the growth of mycelium.

All experiments were carried out in triplicates, and the resulting fungal growth were analyzed as mean ± standard error (*n* = 4).

### 2.6. Microscopic Observation of Fungi Treated with Zinc Nanoparticles

To evaluate the efficacy of ZnNPs for the growth of mycelia, fungal mass was collected by hand-picking using sterilized cork borer from the surface of petri plates containing *D. seriata* culture developed on PDA media treated with 50 ppm of synthesized *E. globules* ZnNPs. Concurrently, the fungal plates treated with tap water were preserved as control samples and observed for the fungal morphology under florescence microscope [40]. 

### 2.7. Scanning Electron Microscopy (SEM) of Fungal Mycelium Treated with Zinc Nanoparticles

To study the effect of ZnNPs on the test fungi, *D. seriata* hyphae was grown on PDA supplemented with 50 ppm of synthesized *E. globules* ZnNPs, and incubated for seven days. The specimen was washed with distilled water following dehydration in a graded ethanol arrangement of up to 100%. SEM studies of treated and untreated test fungi were placed on a carbon-coated copper grid at 80kV by using scanning electron (JSM 5600, JEOL, Tokyo, Japan).

## 3. Results and Discussion

### 3.1. Formation of Zinc Nanoparticles

In the present study, the Zinc oxide was synthesized by using *E. globules* leaf extracts. The change of color to pale yellow from colorless confirmed the formation of ZnNPs (Figure 2). There are reports of other metals where color change demonstrates the preliminary confirmation for the formation of the respective nanoparticles [3,39,41]. The reaction mixture color alteration owing to surface plasmon resonance substantiates the occurrence of ZnNPs [42]. The present observations were established by the previous studies of Shekhawat et al. [43]. The present study indicates the occurrence of rapid reaction of plant derived nanoparticles at room temperature without any additives or reactants. This method is easy and is most suitable for testing the biological activities when compared to other methods, such as physical, chemical, biological or hybrid methods where an external force is required which may carry toxic substances that losses its stability [44].

### 3.2. UV–Visible Analysis of Zinc Nanoparticles 

The synthesis of zinc and zinc oxide nanoparticles has been previously characterized by UV–visible spectroscopy. In this study, UV-Vis spectra of ZnNPs were done by using *E. globules* extracts. The aqueous leaf extract at 300 nm recorded absorption peak, which is attributed to the formation of ZnNPs (Figure 3). These results are in agreement with Shekhawat et al. [43]. In an independent study, it was noticed that the UV emission is oriented to the radiative recombination among electrons in the transmission group and the holes in the valence group expressed in ZnO phosphor powders [45].

### 3.3. Scanning Electron Microscopic study of Zinc Nanoparticles

The SEM images of the samples are shown in Figure 4. The SEM images of ZnNPs synthesized by using *E. globules* extract shows that the agglomerations of particles are much more using this method of preparation. The clustered form of nanoparticles confirms the presence of biological debris in the sample. The size and shape of synthesized ZnNPs were reported by transmission electron microscopic studies. The synthesized ZnNPs were mostly spherical in shape with few elongated particles with variation in particle size, ranging 52 to 70 nm (Figure 5). Previously it was reported that *E. globules* extracts, during synthesis act as an active template which prevents the aggregation of synthesized nanoparticles [29].

### 3.4. FT-IR spectrum of Zinc Nanoparticles 

The FT-IR spectra of *E. globules* extracts along with synthesized ZnNPs extract are shown in Figure 6. The absorption pattern at 756, 768 and 833 cm^−1^ represents the aromatic C-H group that are associated with typical mono-substituted benzene ring 1, 4 di-substituted benzene ring and 1,2,3 tri-substituted benzene ring from different components of its essential oil like, Aromandendrenemyrtenal, Borneol, Camphene, Carvacrol, Citronellal, Citronellyl acetate, Cryptone-α-terpenyl acetate may act as a stabilizing agent during the process of ZnNPs [29]. The absorption spectra at 450–560 cm^−1^ describe the detection of Zn nanoparticle. The peak at 1050 cm^−1^ corresponds to stretching vibration due to C-N amine group. The aromatic ring is present in the absorption region between 1600–1490 cm^−1^; while, the occurrence of aromatic aldehydes was noticed at 2900–2750 cm^−1^ region. The peak at 2049 cm^−1^ was owing to C=C indicating the stretching vibration at that region. The region between 3237–3565 cm^-1^ indicates the presence of O-H stretching bond. Overall, the interpretation of FTIR spectra from our study is well supported by the previous study [29], in which, the synthesis of ZnNPs was performed using leaves of *Corriandrum sativum*.

### 3.5. Antifungal Assessment of Zinc Nanoparticles Against Pathogenic Fungi

Zinc ion solution has showed fewer toxic effects against tested pathogenic fungi, but can be considered as an active biocidal metal (Table 1). The metal solution recorded 13.4 and 13.1% inhibition of *B. dothidea* growth at 50 and 100 ppm, respectively. While the fungus *D.seriata* was maximum inhibited (17.9%) at higher concentration of 100 ppm. Interestingly, the mycelia growth of *A. mali* was reduced in all the tested concentrations (Table 1). In the case of *E. globules* leaf extract alone, a fair to good inhibition rate was recorded against the tested fungi. On the other hand, the *Eucalyptus* derived ZnNPs at 100 ppm showed remarkable inhibition rate of 76.7% for *A.mali,* followed by 65.4 and 55.2% inhibition rate for *B. dothidea* and *D. seriata*, respectively (Table 2). In ZnNPs plus *E. globules* treated plates at 50 and 100 ppm showed good inhibition of the tested fungi. *Eucalyptus* extracts have been previously reported to show anti-microbial activity against *Candida albicans, Trichophyton mentagrophytes, Aspergillus flavus* and *Aspergillus niger* [46]. This shows that Zinc (Zn) was more active against the latter fungus. The anti-microbial potentialities of MgO, ZnO and CaO metal oxide particles have been studied against *E. coli*, *Staphylococcus aureus* and other fungi [47]. The mechanism behind anti-bacterial potential could be the active oxygen species generated by metal oxide particles.

ZnNPs are recognized as multi-functional inorganic nanoparticles with potent anti-bacterial properties. ZnNPs synthesized using *Eucalyptus globules* extract showed an increase in the rate of their antifungal activity over Zn bulk material. The activity of ZnNPs alone was dose- dependent, a fair to good inhibition was observed at 25 ppm, thereafter a significant increase in the inhibition of tested pathogens was recorded at higher concentration of 50 and 100 ppm, respectively as mentioned in Table 2. Our study is in agreement with the research work conducted by [12], who observed a well-defined antifungal ability of bulk ZnNPs solution, which was less corresponding ZnNPs used in their study. According to their study, green ZnNPs showed a remarkable increase in the biological activity against diverse pathogens in comparison to synthetic ZnNPs. The antifungal activity of ZnNPs against *Trichophyton mentagrophyte*, *Microsporum canis*, *Candida albicans* and *Aspergillus fumigatus* was reported by Eman et al. [48]. Their study revealed a largest inhibition in the germination of all the tested fungi at a concentration of 40 mg/mL. However, there are no previous reports regarding nano-fungicidal study, especially ZnNPs for the phytopathogenic fungi on fruit crops like apple. Furthermore, the synergetic influence of ZnNPs and *Eucalyptus globules* extracts with equal proportionate was evaluated on the mycelial growth of fungi. Due to the synergetic action, a complete inhibition (100%) of the mycelium was noticed at 100 ppm in case of *B. dothidea* and *A. mali.* As discussed early, *E. globules* extract has certain levels of fungicidal activity against the test fungi, which was added to the total impact of ZnNPs, which inhibits the mycelial growth. In this way, the entirety of the inhibition effect was expressed when both the samples were equally mixed.

### 3.6. Effect of Zinc Nanoparticles on Fungal Hyphae

#### 3.6.1. Microscopic Observation of Fungi Treated with Zinc Nanoparticles

The rupture at hyphae tip, the region for the formation of new conidia, along with disconnected conidia was detected concurrently in two fungi with microscopic study (Figure 7). The release of cellular materials might be due to the damage on the surface of fungal hyphae which is caused during hyphal contraction. On the other hand, the water-treated hyphae stay intact without any hyphal damage. 

#### 3.6.2. Scanning Electron Microscopy (SEM) Study of Fungal Mycelia upon ZnNPs Treatment 

The microscopic observation of the impact of nanoparticles on growing hyphae revealed that ZnNPs clearly damaged hyphae of *D. seriata* (Figure 8), while hyphae treated with water appeared to remain intact. In the treatment of nanoparticles, the structure and shape of hyphal walls turned abnormal by the deformations and damages of the *D. seriata* hyphal wall membrane, deterioration of reproductive structures resulting severe broken hyphal wall layers remaining few with less damage showed shrunken hyphae (Figure 8). Interestingly, these modifications in the mycelium structures did not influence any changes in the life cycle of the fungus. A similar observation was reported by Villamizar-Gallardo et al. [49] that the synthesized AgNPs produce significant structural damages of *Aspergillus flavus, but* did not affect the formation of the life cycle of the fungus.

## 4. Conclusions

The present investigation reports the green and eco-friendly synthesis of ZnNPs by *Eucalyptus globules* leaf extract as reducing agent. Change in color from colorless to pale yellow was due to the phenomenon of resonance plasmon absorption transitions, which is considered as the primary indication of ZnNPs formation. The consequence of additional characterization process like SEM and TEM demonstrated that the ZnNPs were spherical or globular shaped with the undefined varied size of 52-70 nm. The phytochemicals of *E. globules* were consistent for the fabrication of ZnNPs as demonstrated by FTIR results. A further study provided the evidence for the substantial fungicidal property of ZnNPs against *D. seriata, B. dothidea* and *A. mali,* highly economically significant phytopathogenic fungi infecting apple orchards. At 100 ppm concentration, the ZnNPs showed 65.4, 55.2 and 76.7% inhibition of growth rate against *B. dothidea, D. seriata* and *A. mali*, respectively. The microscopic observation of fungal growth of treated with ZnNPs revealed that synthesized ZnNPs damage the surface of the fungal hyphae, thereby discharging cellular materials, ensuing in the contraction of hyphae as confirmed by the SEM study. Hence, the current study describes that ZnNPs were synthesized by green, facile and easy method. The outstanding antifungal property of ZnNPs against phytopathogenic fungi of apple orchards will have a significant, and a beneficial, impact of utilizing nanoparticles for the fungal pest management and enhanced plant protection in fruit crops.

## Figures and Tables

**Figure 1 biomolecules-10-00425-f001:**
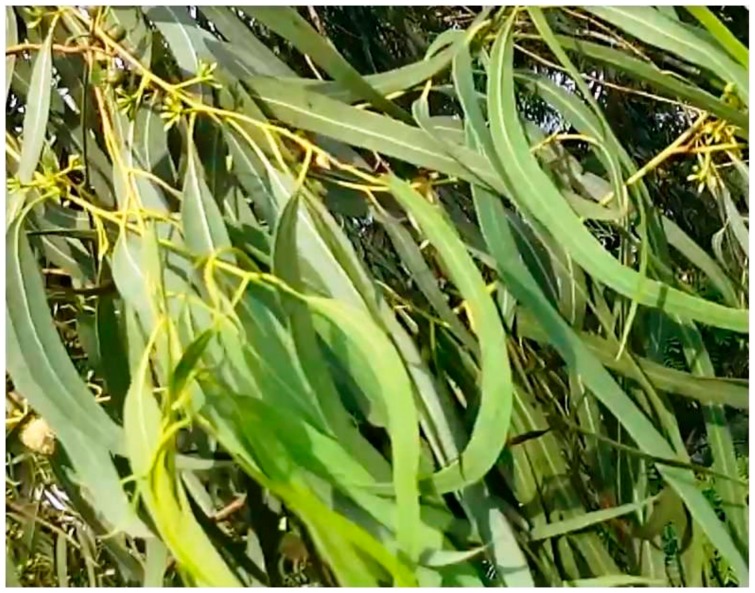
*Eucalyptus globules* leaves, collected from Jamin Pallavarm Vels University, Chennai used for the synthesis of nanoparticles.

**Figure 2 biomolecules-10-00425-f002:**
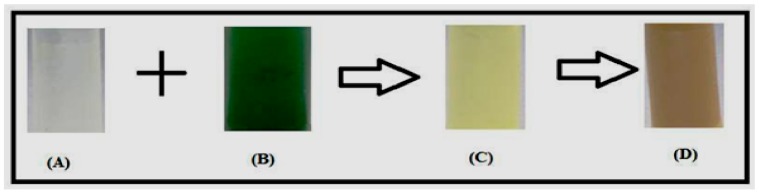
Formation of Zinc nanoparticles; (**A**) Eucalyptus broth, (**B**) one mM Zinc nitrate solution, (**C**) Reaction mixture and (**D**) Zinc nanoparticle solution.

**Figure 3 biomolecules-10-00425-f003:**
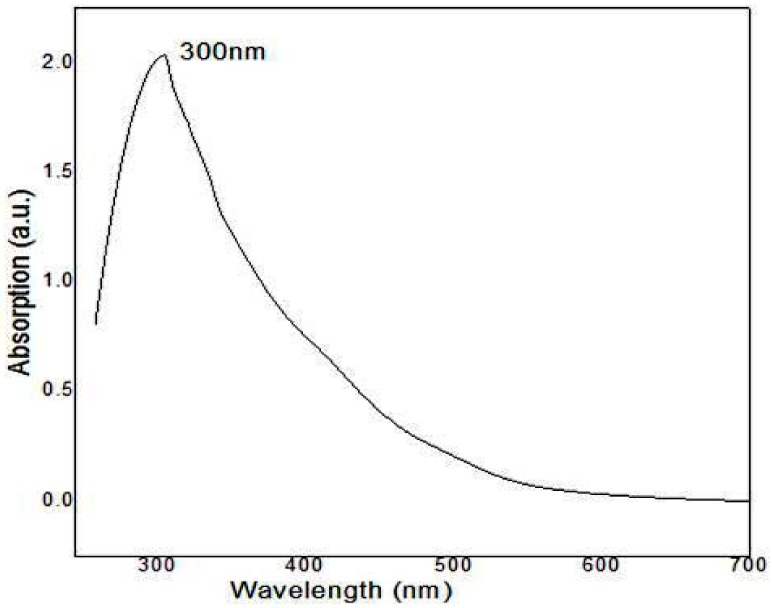
UV-visible spectroscopy of zinc nanoparticles synthesized using *Eucalyptus globules* leaf extract showing absorption peak at 300 nm.

**Figure 4 biomolecules-10-00425-f004:**
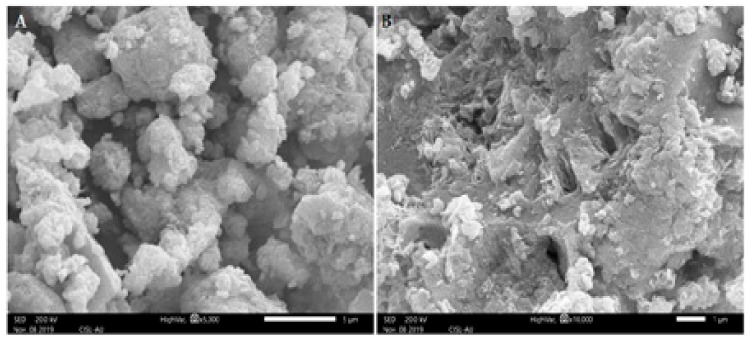
SEM images of zinc nanoparticles synthesized using *Eucalyptus globules* leaf extract. Bar scale—5 µm (**A**) and 1 µm (**B**).

**Figure 5 biomolecules-10-00425-f005:**
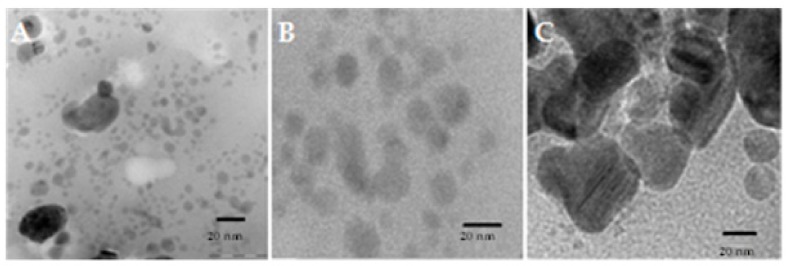
Images of Transmission electron microscopy (TEM) of synthesized ZnNPs (**A**–**C**) showing spherical or globular (**B**,**C**) shaped with the undefined varied size of 52-70 nm. Bar scale—20 nm.

**Figure 6 biomolecules-10-00425-f006:**
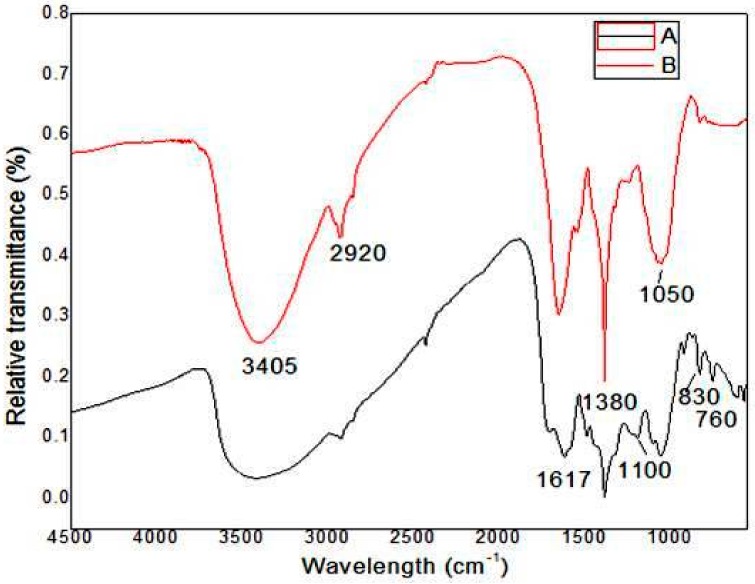
Fourier transform infrared (FTIR) spectra of (A) *Eucalyptus globules* leaf extract, (B) zinc nanoparticles expressing relative transmittance (%) of functional groups.

**Figure 7 biomolecules-10-00425-f007:**
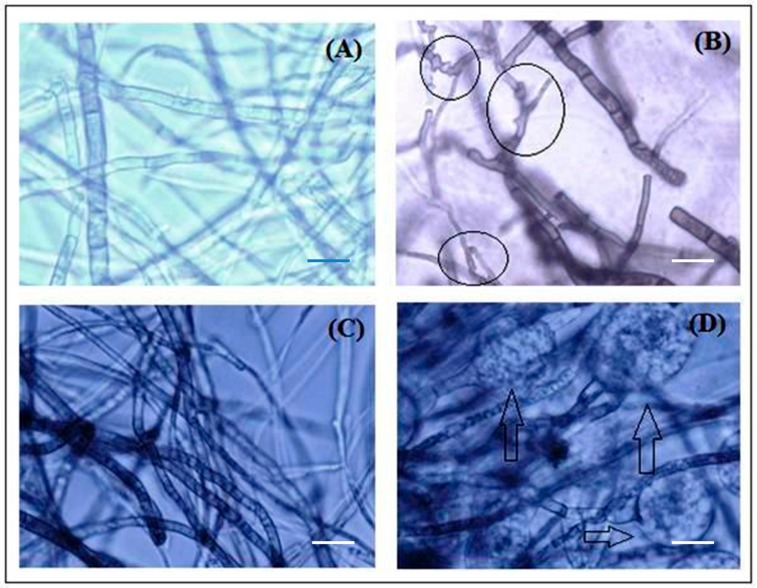
Impact of biosynthesized zinc nanoparticles on the morphology of fungi before and after treatment observed in the light microscope with 40× magnification. (**A**), (**B**) is *Diplodia seriata* and (**C**), (**D**) is *Botryosphaeria dothide;* While as, (**A**), (**C**) shows untreated fungi, (**B**), (**D**) shows cytoplasmic aberration caused by zinc nanoparticles. Bar scale—5µm.

**Figure 8 biomolecules-10-00425-f008:**
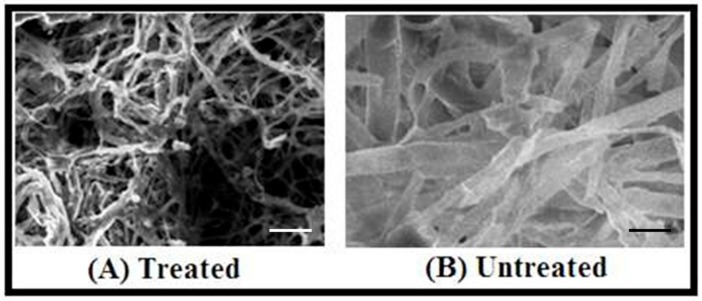
Images of scanning electron microscopy (SEM); (**A**) (treated): Damaged hyphae of *Diplodia seriata due to* treatment with ZnNPs, (**B**) (untreated): Showing normal mycelial growth of *Diplodia seriata*. Bar scale—5 µm.

**Table 1 biomolecules-10-00425-t001:** In vitro inhibitory rate (%) of phytopathogenic fungi using different concentrations of *Eucalyptus globulus* leaf extract and Zn ion solution.

**Test Fungi**	**Inhibition rate (%)**							
	*Eucalyptus globulus*				Zn ion solution			
	10 ppm	25 ppm	50 ppm	100 ppm	10 ppm	25 ppm	50 ppm	100 ppm
*Botryosphaeria dothidea*	10.1 ± 0.3 ^c^	18.1 ± 1.3 ^b^	22.4 ± 0.4 ^b^	27.3 ± 2.4 ^b^	2.2 ± 0.4 ^b^	9.5 ± 1.6 ^c^	13.4 ± 2.5 ^b^	13.1 ± 0.7^a^
*Diplodia seriata*	7.2 ± 2.4 ^b^	10.5 ± 2.1 ^c^	11.8 ± 0.2 ^c^	21.8 ± 2.1^bc^	5.2 ± 0.5 ^b^	12.3 ± 2.5 ^b^	14.4 ± 1.6 ^b^	17.9 ± 0.5 ^b^
*Alternaria mali*	40.9 ± 1.4^a^	49.2 ± 2.1 ^a^	62.82 ± 1.2 ^a^	78.7 ± 2.4 ^a^	22.3 ± 1.2 ^a^	32.8 ± 1.4 ^a^	36.3 ± 0.5 ^a^	35.7 ± 2.5 ^a^

The inhibition rates in (%) are the means of four individual replicates ± standard errors (SEs). The letter(s) on the means in each column are not significantly different as per Tukey’s HSD analysis (SPSS 20.0, SPSS, Inc. Chicago, IL, USA).

**Table 2 biomolecules-10-00425-t002:** In vitro inhibitory rate (%) of phytopathogenic fungi using different concentrations of synthesized ZnNPs alone and synergistic ZnNPs plus *Eucalyptus globulus* leaf extract.

**Test Fungi**	**Inhibition Rate (%)**							
	ZnNPs				ZnNPs+ *Eucalyptus Globulus*			
	10 ppm	25 ppm	50 ppm	100 ppm	10 ppm	25 ppm	50 ppm	100 ppm
*Botryosphaeria dothidea*	21.1 ± 0.2 ^b^	46.3 ± 3.2 ^ab^	60.5 ± 0.4 ^b^	65.4 ± 0.6 ^a^	27.1 ± 1.4 ^b^	50.3 ± 0.2 ^a^	61.5 ± 1.3 ^a^	67.1 ± 2.5 ^b^
*Diplodia seriata*	16.5 ± 0.4 ^bc^	31.3 ± 1.3 ^c^	53.1 ± 0.1^bc^	55.2 ± 1.2 ^c^	36.5 ± 1.2 ^b^	48.3 ± 1.2 ^ab^	65.3 ± 0.8 ^a^	66.7 ± 1.3 ^b^
*Alternaria mali*	36.6 ± 1.4 ^a^	51.6 ± 1.5 ^a^	73.5 ± 1.4 ^a^	76.7 ± 1.4 ^a^	20.6 ± 0.3^bc^	39.6 ± 0.4 ^c^	53.3 ± 1.4 ^b^	72 ± 1.9^a^

The inhibition rates in (%) are the means of four individual replicates ± standard errors(SEs). The letter(s) on the means in each column are not significantly different as per Tukey’s HSD analysis (SPSS 20.0, SPSS, Inc. Chicago, IL, USA).

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
