# Peer review of "Green Synthesis and Characterization of Zinc Oxide Nanoparticles Using Eucalyptus globules and Their Fungicidal Ability Against Pathogenic Fungi of Apple Orchards"

_biomolecules, 2020, doi:10.3390/biom10030425_

Round 1
Reviewer 1 Report
The article highlights the advantages of producing ZnNPs with antifungal activity using Eucalyptus globules leaf extract. However, in vitro studies presented promising results, the following revisions should be taken in attention:
- Despite the manuscript is written concisely, there are some grammatical errors through the paper (e.g. line 55 “have also widely been studied”, lines 56, 63 and 65 “ZnNPs nanoparticles”, line 59 “potential of Zinc oxide nanoparticles was observed to be effective than that of ZnNPs”,….). Attention should also be given to the use of abbreviations (e.g. line 99 “Zn nanoparticles” should be ZnNPs, line 124 “PDA” what is it means?).
- In the abstract, the authors said that the size of ZnNPs range between 52-70 nm, but in the rest of the text, including in the TEM images, they mention that is around 52 nm. Moreover, in line 184 it is mentioned that “variation in particle size, ranging 40 to 80 nm”.
- The authors claim a green approach, but they use mercuric chloride to sterilize. Why did they use this toxic chemical compound?
- In Figure 3, the authors claim that the absorption peak is at 300 nm, but considering the x-axis is higher than that.
- The SEM and TEM images of Zn nanoparticles have poor resolution.
- Instead of tables 1a and 1b, should be tables 1 and 2, or at least “a” and “b” should be included in the tables heading.
- In lines 238 and 239, the authors mention percentages higher than those present in the tables.
- In conclusion, the authors mention XRD analysis that was not performed in this study. Moreover, the percentages of inhibition of growth rate are higher than what was obtained. Consequently, conclusions should be carefully revised.
Author Response
Reviewer 1:
The article highlights the advantages of producing ZnNPs with antifungal activity using Eucalyptus globules leaf extract. However, in vitro studies presented promising results, the following revisions should be taken in attention:
Response: We would like to express our special thanks to the Reviewer for his/her critical observation and provided constructive comments that have helped us improve the quality of our manuscript. We have taken all our efforts to revise the manuscript, taking into account all the comments and suggestions of the Reviewer.
Comment 1: Despite the manuscript is written concisely, there are some grammatical errors through the paper (e.g. line 55 “have also widely been studied”, lines 56, 63 and 65 “ZnNPs nanoparticles”, line 59 “potential of Zinc oxide nanoparticles was observed to be effective than that of ZnNPs”,….). Attention should also be given to the use of abbreviations (e.g. line 99 “Zn nanoparticles” should be ZnNPs, line 124 “PDA” what is it means?).
Response: We thank the Reviewer for this critical observation. All the topographical errors are now fixed in the revised manuscript (L.62, 127, 141, 146, 170, 176, 252, 305, 306 and 318).
Comment 2: In the abstract, the authors said that the size of ZnNPs range between 52-70 nm, but in the rest of the text, including in the TEM images, they mention that is around 52 nm. Moreover, in line 184 it is mentioned that “variation in particle size, ranging 40 to 80 nm”.
Response: We apologize for this topographical mistake/s. The size of the ZnNPs ranged between 52-70 nm. The same values are now fixed in the revised manuscript (L.34, 220, 231 and 341).
Comment 3: The authors claim a green approach, but they use mercuric chloride to sterilize. Why did they use this toxic chemical compound?
Response: Thank you very much for this comment and critical observation. Yes, we agree that mercuric chloride is toxic chemical. In this investigation, we have use low concentration of 0.1% chemical and wash repeatedly washed (four – five times with sterile distilled water) to ensure the chemical residue (if any) is completely washed (L.106). In the further, we will use sodium hypochlorite of surface sterilization of plant materials.
Comment 4: In Figure 3, the authors claim that the absorption peak is at 300 nm, but considering the x-axis is higher than that.
Response: Thank you very much for this inquiry. Actually, the absorbance was read between 200-700 nm using spectrophotometer (UV-260, Shimadzu), the peak absorbance was recorded at 300 nm and it was displayed and labeled by the system in the Figure 3 (P.7; L.211 and 212).
In addition, we have compared with the chemical Zinc nanoparticles in the rage of 200-900 nm and the results are absolutely matching with the absorbance peak at 300nm (FOR REVIEWER ONLY - Below Figure). Since, this is not the objective of our present investigation we are extremely sorry for not incorporating this Figure in the revised manuscript.
Comment 5: The SEM and TEM images of Zn nanoparticles have poor resolution.
Response: We highly appreciate the Reviewer for his/her critical observation of our manuscript. As suggested by the Reviewer, we have now supplied a high resolution of SEM (Figure 4) (P.7; L.226 and 227) and Tem (Figure 5) in the revised manuscript (P.8; L.230 and 231).
Comment 6: Instead of tables 1a and 1b, should be tables 1 and 2, or at least “a” and “b” should be included in the tables heading.
Response: We thank the Reviewer again for this comment and suggestion. To avoid confusion, the Table 1a is renamed as “Table1” and Table 1b as “Table 2” (P.9 and 10).
Comment 7: In lines 238 and 239, the authors mention percentages higher than those present in the tables.
Response: The authors would like to thank the Reviewer for this critical observation and comment. Sorry, for these topographical errors, we have now fixed all these errors and revised the sentences in accordance with our obtained results (L.286-289).
Comment 8: In conclusion, the authors mention XRD analysis that was not performed in this study. Moreover, the percentages of inhibition of growth rate are higher than what was obtained. Consequently, conclusions should be carefully revised.
Response: Thank you so much for raising this query that is interconnected with above comment. We sincerely apologize for wrong interpretation of the text/ data, we have now fixed all these errors and revised the sentences in accordance with our obtained results (L.340, 345).

Reviewer 2 Report
This study investigates the kinetics formation of Zn-nanoparticles using Eucalyptus globules as reducing agent. This approach is aimed to take advantage of the antifungal properties of Eucalyptus globules in order to justify the fungicidal properties of Zn-nanoparticles against phytopathogenesis of apple orchards.
Zn-nanoparticles are characterized by means of various spectroscopical techniques beside SEM and TEM microscopies and an accurate analysis on assessment of Zn-nanoparticles has been carried out.
My opinion is that this manuscript correctly describes the results that have been obtained, which, however, are of rather limited interest from a general point of view but may be of some interest from a strictly applicative point of view.
I wonder if this manuscript might be of interest to the readers of this Journal or if perhaps it could find a better place in a more specialized Journal.
Minor points:
the technical quality of Figure 2 is rather poor. It should be improved
The result shown in Fig. 8 is described too succinctly. The authors should say in more detail what it means.
Line 230 - Table 1 should be Table 2
Author Response
Reviewer 2:
This study investigates the kinetics formation of Zn-nanoparticles using Eucalyptus globules as reducing agent. This approach is aimed to take advantage of the antifungal properties of Eucalyptus globules in order to justify the fungicidal properties of Zn-nanoparticles against phytopathogenesis of apple orchards. Zn-nanoparticles are characterized by means of various spectroscopical techniques beside SEM and TEM microscopies and an accurate analysis on assessment of Zn-nanoparticles has been carried out.
Response: We are very glad that the Reviewer has positively evaluated our manuscript, and provided constructive comments and suggestions that have helped us improve the quality of our manuscript.
My opinion is that this manuscript correctly describes the results that have been obtained, which, however, are of rather limited interest from a general point of view but may be of some interest from a strictly applicative point of view. I wonder if this manuscript might be of interest to the readers of this Journal or if perhaps it could find a better place in a more specialized Journal.
Response: The authors would like to thank the Reviewer for his/her opinion and remarks on our manuscript. The present investigation deals with synthesized ZnNPs from E. globules which exhibit a remarkable impact on exploitation of synthesized ZnNPs for fungal pest management and ensure protection in fruit crops. Furthermore, the manuscript is submitted to a special issue entitled “Plant-based green synthesis of nanoparticles: production, characterization and applications” of Biomolecules Journal.
https://www.mdpi.com/journal/biomolecules/special_issues/green_synthesis_nanoparticles
Thus, we believe that the content described in our manuscript falls within the scope of the journal Biomolecules.
Minor points:
Comment 1: The technical quality of Figure 2 is rather poor. It should be improved.
Response: Thank you very much for this important advice and suggestion. As suggested by the Reviewer, we have now supplied a high resolution Figure 2 (P.6).
Comment 2: The result shown in Fig. 8 is described too succinctly. The authors should say in more detail what it means.
Response: Thank you very much for this comment and suggestion with which we totally agree. The results of section 3.6.2. Scanning Electron Microscopy (SEM) study of fungal mycelia is now elaborated in detail for better understanding (L.321-328).
Comment 3: Line 230 - Table 1 should be Table 2.
Response: We would like to thank the Reviewer for this suggestion. To meet the Reviewer’s suggestions, Table 1b is renamed as Table 2 (L.277).
Reviewer 3 Report
This paper, by Ahmad et. al, studies the ability of Eucalyptus globulus broth to be an efficient reducing agent, leading to the rapid 30 formation of ZnNPs of varied shapes and sizes. This system is well designed and characterized and the rational is also explained. I believe this manuscript can be published after minor revision. Here are some minor comments
- Introduction should include the major drawbacks of currently used commercial production. This will add some importance to the paper.
- Plant extract should be further characterized. At least name the major components of discuses them.
- It will also be a good idea to compare the performance of the suggested method to at least one known methods. This will give the readers some retrospective regarding the efficacy.
- Figure 3: please compare UV spectra to the NP to one of commercially available particles.
- Figure 4 is very vague and hard to follow. Please improve or delete.
Author Response
Reviewer 3:
This paper, by Ahmad et. al, studies the ability of Eucalyptus globulus broth to be an efficient reducing agent, leading to the rapid 30 formation of ZnNPs of varied shapes and sizes. This system is well designed and characterized and the rational is also explained. I believe this manuscript can be published after minor revision. Here are some minor comments.
Response: It is indeed glad to note that the Reviewer highly evaluated our manuscript, and provided constructive comments and suggestions that have helped us improve the quality of the manuscript. We have addressed each of the comment very objectively and the responses for the individual comments are as below:
Comment 1: Introduction should include the major drawbacks of currently used commercial production. This will add some importance to the paper.
Response: We thank the Reviewer again for pointing this out, with which we totally agree. Thus, we have added the following paragraph into the Introduction:
“These methods can be adopted for either chemical synthesis or plant-derived synthesis of nanoparticles. For chemical synthesis of metal nanoparticles a specific external catalysts and synchronized conditions are required, in case of plant-derived nanoparticles catalysts in the form of co-enzymes are secreted by plants which are non-toxic reactants and eco-friendly and the reaction occurs at favorable room temperature conditions. Hence, there is an emerging concern towards the use of plant extracts so called phytosynthesis of nanoparticle due to presence of plant bio-molecules that can act as capping and reducing agents. These agents thus increase the rate of reduction and stabilization of nanoparticles. In addition, the use of plants is an adventitious process over commercial products because it does not involve the long process of growing cell cultures and is more useful for large-scale nanoparticle synthesis. Hence, an increase in the biogenesis of nanoparticles using plant materials is tremendously growing, for instance the synthesis of ZnNPs by using…………..” (L.70-81).
Comment 2: Plant extract should be further characterized. At least name the major components of discuses them.
Response: Thank you so much for this comment. As suggested by the Reviewer, we have mentioned the major phytoconstituents involved Aromandendrene myrtenal, Borneol, Camphene, Carvacrol, Citronellal, Citronellyl acetate, Cryptone- α- terpenyl acetate secreated by Eucalyptus globules that may act as a stabilizing agent during the process of ZnNPs. The same has been discussed in the Introduction (L.92-94) and Results/Discussion section (L.237-239).
Comment 3: It will also be a good idea to compare the performance of the suggested method to at least one known methods. This will give the readers some retrospective regarding the efficacy.
Response: We appreciate the Reviewer for his/her critical observation of our manuscript. As suggested by the Reviewer we have compared the present method of synthesis with other methods with citation in the results/discussion section as below:
“The present study indicates the occurrence of rapid reaction of plant derived nanoparticles at room temperature without any additives or reactants. This method is easy and is most suitable for testing the biological activities when compared to other methods such as physical, chemical, biological or hybrid methods where a external force is required which may carry toxic substances that losses its stability [44]. (L.190-194).
- Jeevanandam, J.; Barhoum, A.; Chan, Y.S.; Dufresne, A.; Danquah, M.K. Review on nanoparticles and nanostructured materials: history, sources, toxicity and regulations. Beilstein J Nanotechnol. 2018, 9, 1050–1074.
Comment 4: Figure 3: please compare UV spectra to the NP to one of commercially available particles.
Response: Thank you so much for this suggestion. Yes, we have compared with the chemical Zinc nanoparticles in the rage of 200-900 nm and the results are absolutely matching with the absorbance peak at 300nm (FOR REVIEWER ONLY - Below Figure). Since, this is not the objective of our present investigation we are extremely sorry for not incorporating this Figure in the revised manuscript.
Comment 5: Figure 4 is very vague and hard to follow. Please improve or delete.
Response: Thank you very much for this important advice and suggestion. As suggested by the Reviewer, we have now supplied a high resolution of SEM (Figure 4) (P.7; L.226-227).

Round 2
Reviewer 3 Report
Authors fully dressed my concerns.